# Harnessing Genomics of *Diaporthe amygdali* for Improved Control of Peach Twig Canker and Shoot Blight (TCSB)

**DOI:** 10.3390/plants14192960

**Published:** 2025-09-24

**Authors:** Silvia Turco, Federico Brugneti, Antonella Cardacino, Angelo Mazzaglia

**Affiliations:** Department of Agriculture and Forest Sciences, University of Tuscia, 01100 Viterbo, Italy; federico.brugneti@unitus.it (F.B.); antonella.cardacino@unitus.it (A.C.)

**Keywords:** *Diaporthe amygdali*, *Prunus persica*, Twig Canker and Shoot Blight (TCSB), whole-genome sequencing, hybrid de novo assembly

## Abstract

In recent years, symptoms of Twig Cankers and Shoot Blight (TCSB) have re-emerged in several Italian peach orchards, particularly within key production areas of the Emilia-Romagna region. The fungal pathogen *Diaporthe amygdali* is recognized as the primary causal agent of TCSB, leading to the rapid desiccation of shoots, flowers, leaves, and branches, often accompanied by resin exudation from cankers that appear in late winter or early spring. Given Italy’s position as the second-largest peach producer in Europe, ensuring sustainable yields and high fruit quality necessitates a deeper understanding of *D. amygdali* biology and the development of effective diagnostic and management tools. This study employed a hybrid whole-genome sequencing strategy, combining Illumina short-read and PacBio long-read technologies, to generate the first high-quality genome assembly of *D. amygdali* isolated from peach. The genome analysis revealed candidate virulence genes and other factors involved in pathogenicity, deepening our understanding of the infection strategies employed by *D. amygdali*. These findings may support the potential development of sustainable, effective strategies against TCSB, ultimately supporting resilient peach production in Italy and beyond.

## 1. Introduction

Fungi belonging to the genus *Diaporthe* (family Diaporthaceae, order Diaporthales) represent a large group of microorganisms with endophytic, saprobic, and pathogenic lifestyles across a wide range of plant hosts [1]. Notably, several species within this genus are recognized as significant plant pathogens, responsible for severe diseases in economically important crops [2]. Among them, *Diaporthe amygdali* is a well-known pathogen associated with twig canker in almonds (*Prunus dulcis* Mill.) and peach (*Prunus persica* (L.) Batsch), causing economic losses in the main producing regions [3,4]. This species was initially described as *Fusicoccum amygdali* in France over a century ago by Delacroix [5] and subsequently in Italy on almonds by Canonaco [6]. For many decades, it remained classified under this name. It was closely associated with its anamorphic form *Phomopsis amygdali,* which further complicated its taxonomy due to morphological similarities with other members of the *Diaporthe* species complex [7,8]. The introduction of molecular techniques and multilocus phylogenetic analyses prompted a comprehensive taxonomic revision. Sequencing of loci such as the Internal Transcribed Spacer (*ITS*) region of the nuclear rDNA, the translation elongation factor 1-α (*tef-1α*), β-tubulin (*tub*), and calmodulin (*cal*) genes provided robust phylogenetic evidence for its reclassification into the genus *Diaporthe* [9].

*D. amygdali* has historically been reported in southern European countries such as Italy, Greece, and France, particularly in regions with Mediterranean climates [10,11]. Recent studies have confirmed its continued presence and impact in these areas, with additional reports from Spain [12,13] and Portugal [7]. However, the earliest documented reports likely date back to 1989 [14]. In recent decades, *D. amygdali* re-emerged as a pathogenic agent of peach by causing Twig Canker and Shoot Blight (TCSB), which affects young branches and shoots, resulting in necrotic lesions, dieback, and, in severe cases, the decline or death of plants. Infected plants also show exudate leakage from cankers throughout the growing season, potentially attributed to fungal toxins, such as fusicoccin A, a compound that disrupts host cellular function and contributes significantly to symptom development [4,15]. The impact of the disease has been significant in Italy, which is the second-largest EU producer of peaches, with a total production of 744,452.5 t in 2023 [16]. In particular, the main symptoms (rapid desiccation of shoots, flowers, leaves, and branches) have been observed in peach orchards in Emilia-Romagna, one of the most important production regions, which experienced serious production losses [4,17].

Given the severe impact of Twig Canker and Shoot Blight (TCSB) on peach orchards, particularly in key production areas, the development of effective disease management strategies is crucial. In this context, the accurate identification and effective management of plant pathogens remain critical challenges for sustainable agriculture and global food security [18]. In this regard, genomics has emerged as a powerful tool for advancing understanding of pathogen biology, unraveling plant–pathogen interactions and enabling precise control [19,20,21,22]. It also provides comprehensive insights into the genetic makeup of plant pathogens, facilitates species- and strain-level identification, mitigating the misclassification errors [23]. In addition, beyond taxonomy, genomics enable the identification of key genes involved in pathogenicity, virulence, and host specificity, offering essential information for the development of targeted control strategies, including resistant crop varieties and specific biocontrol agents [24,25,26,27]. Moreover, the detection of genes associated with resistance to fungicides or other antimicrobial compounds facilitates early intervention and adaptation of management practices, thereby mitigating the risk of treatment failure [28,29,30].

Despite the growing application of genomic approaches in plant pathology, no genomic data are currently available for *D. amygdali* isolates infecting peach, limiting our understanding of its pathogenic mechanisms and potential targets for control. To fill this gap, the present study provides the first comprehensive genomic and comparative analysis of the *D. amygdali* isolate DA-1, representing a key step toward improving species identification, elucidating virulence patterns, and supporting the development of more effective and sustainable disease management strategies.

## 2. Results

### 2.1. Assembly Metrics and Genomic Overview

High-molecular-weight (HMW) DNA was sequenced using a hybrid approach. Illumina short-read sequencing yielded 3.96 Gb of paired-end data, with Q20 and Q30 scores exceeding 99.24% and 96.31%, respectively. PacBio HiFi long-read sequencing generated 487,677 circular consensus sequencing (CCS) reads, with an average read length of 8937 bp and a maximum read length of 44,595 bp. Genome assembly was performed using Hifiasm with PacBio long reads, followed by polishing with Polca using Illumina reads. The final draft genome consisted of 14 contigs, totaling 51.43 Mb, with an N50 of 6.8 Mb, a GC content of 52.64%, and an overall sequencing coverage of 76.42× (Figure 1, Table 1). Notably, five nuclear contigs span from telomere-to-telomere, while four carry a single telomere, suggesting that they represent chromosome arms or near-complete chromosomes. These findings indicate the presence of up to seven nuclear chromosomes. The tenth contig represents the complete circular mitochondrial genome. The remaining contigs include unassembled regions, such as ribosomal RNA genes and retrotransposable elements. Gene prediction using EvidenceModeler (EVM) integrated multiple ab initio and evidence-based algorithms, resulting in the annotation of 15,184 nuclear genes and 22 mitochondrial genes, totaling 15,206 predicted genes, along with 51 rRNAs, 160 tRNAs, and 84 ncRNAs. Repeat annotation identified 663 LTR/Gypsy retroelements, covering 1,733,022 bp (3.37% of the genome), as well as 24 hobo-Activator DNA transposons (10,162 bp, 0.02%) and 13 Tourist/Harbinger elements (2898 bp, 0.01%). The assembled genome has been deposited in the NCBI Genome database under the accession number JBPXNF000000000.

### 2.2. Functional Annotation

The predicted gene set of *D. amygdali* underwent extensive functional annotation using multiple public databases. Gene Ontology (GO) terms were assigned to 7917 genes, providing insights into associated biological processes, molecular functions, and cellular components (Appendix A—GO Annotation). KEGG pathway analysis identified 3637 genes involved in diverse metabolic and signaling pathways (Figure 2, Appendix A—KEGG Annotation). In parallel, KOG annotations were retrieved for 6857 genes, reflecting conserved orthologous functions across eukaryotes (Appendix A—KOG Annotation). Pfam domain analysis assigned functional protein families to 10,426 genes (Appendix A—Pfam Annotation). SwissProt homology searches returned matches for 8253 genes, while the broader TrEMBL database matched 13,569 genes, capturing both reviewed and unreviewed protein sequences (Appendix A—SwissProt & TrEMBL Annotations). Collectively, these annotations cover a substantial portion of the predicted gene set, supporting high-quality gene prediction and offering a solid foundation for downstream functional, evolutionary, and comparative genomic studies.

### 2.3. AT-Rich Regions

The *D. amygdali* genome exhibited a bimodal GC content distribution, suggesting compartmentalization into distinct genomic regions. Approximately 1.12% of the genome was classified as AT-rich regions (R0), with an average length of 4.91 Kbp and a GC content peak of 29.4%. In contrast, the remaining 98.9% corresponded to GC-rich regions (R1), averaging 365 Kbp in length and peaking at 53.4% GC content, with a gene density of 295 genes per Mbp (Figure 3A). Notably, no genes were detected in the AT-rich R0 regions. The bimodal GC distribution is consistent with Repeat-Induced Point mutation (RIP) activity—a genome defense mechanism in fungi that mutates repetitive sequences by converting cytosines to thymines, thereby silencing transposable elements [31]. To investigate RIP, dinucleotide frequencies were assessed using RIPCAL. The R0 regions were enriched in AA, AT, TT, and TA dinucleotides relative to R1 (Figure 3B), consistent with RIP-induced bias. The RIP product index (TpA/ApT) was 1.26, and the RIP substrate index [(CpA + TpG)/(ApC + GpT)] was 0.63, both indicative of active RIP. Conversely, the R1 regions showed no evidence of RIP, with a TpA/ApT ratio of 0.65 and a substrate index of 1.28, reflecting preserved GC-rich sequences. These findings support a model in which RIP contributes to the formation of gene-poor, AT-rich compartments (R0), while gene-dense GC-rich regions (R1) remain unaffected. Finally, a BLASTn search for rid alleles, which are essential for RIP activity [32], revealed no homologs in the *D. amygdali* genome, suggesting that the observed RIP signature may result from ancestral activity or divergent RIP machinery.

### 2.4. Secondary Metabolite Biosynthetic Gene Clusters

AntiSMASH analysis revealed a wide array of biosynthetic gene clusters (BGCs) associated with the production of secondary metabolites, many of which are implicated in pathogenicity and host interactions. High-similarity clusters included those responsible for the synthesis of choline (type I non-ribosomal peptide synthetase-like, NRPS-like), as well as the type I polyketide synthases (T1PKSs) nectriapyrone, BAB/BAA, and alternariol. Additional clusters identified with medium similarity included T1PKS-associated neosartorin, scytalone, burnettiene, alternapyrone, and the terpene copalyl diphosphate. Several more clusters showed low similarity to known BGCs, including those for zearalenone, hexadehydroastechrome, cryptosporioptide B, brassicicene C, flavoglaucin, squalestatin S1, 3′-methoxy-1,2-dehydropenicillide, chaetoviridin E, fusarielin H, tricholignan A, patulin, HEx-PKS23 polyketide, 4-methyl-5-dimethyltetradecahexaene-β-ketolactone, lijiquinone, betaenone A, PR-toxin, and depudecin. This diverse repertoire of secondary metabolite BGCs underscores the metabolic versatility of *D. amygdali* and suggests a strong potential for the production of phytotoxic compounds that may contribute to host colonization and virulence.

### 2.5. Identification of Features Related to Host Invasion and Colonization

The first barrier against fungal pathogens is the plant cell wall, which is primarily composed of three classes of polysaccharides: cellulose, hemicelluloses, and pectins. For a successful invasion, pathogens must overcome this physical defense by degrading its components. To facilitate this process, fungi secrete a diverse array of carbohydrate-active enzymes (CAZymes) that target plant cell wall polysaccharides, storage carbohydrates, and other substrates essential for fungal nutrition, deeper described in Section 2.7 [33].

To support colonization, pathogens must also extract nutrients from host tissue. A total of 138 membrane transporters, classified according to the TCDB database, were identified and include proteins associated with mitochondrial respiratory complexes, ATP synthases, and ion pumps (Appendix A). These transporters play critical roles in energy production, ion homeostasis, and metabolite exchange—functions that are essential during host invasion. Among them, several nutrient transporters for sugars, amino acids, and inorganic ions were detected, along with multidrug resistance proteins and efflux pumps likely involved in toxin export. The identification of mitochondrial import machinery and stress response proteins further underscores the role of membrane transport systems in fungal adaptation and pathogenicity. In addition, cytochrome P450 monooxygenases contribute significantly to fungal metabolism. These enzymes participate in the biosynthesis of secondary metabolites, including toxins and pigments, and are involved in intercellular communication, plant–pathogen interactions, sporulation, growth, nutrient acquisition, and stress responses [34]. A total of 834 cytochrome P450s were identified in *D. amygdali* DA-1 (Appendix A), highlighting their potential role in pathogenicity and environmental adaptation.

### 2.6. Comparative Genomics Analysis Among Diaporthe Isolates

The annotated protein-coding genes of the *D. amygdali* DA-1 isolate were compared with those of 13 others annotated *Diaporthe* isolates using OrthoFinder. The analysis identified 18,677 orthogroups, encompassing a total of 208,671 genes, which represents 97.1% of all genes assigned to orthogroups (Figure 4). Among these, 7040 orthogroups were shared across all isolates, and 4344 were single-copy orthogroups, indicating conserved core gene content. In contrast, 661 orthogroups, comprising 2315 genes, were species-specific, suggesting potential roles in host specificity or niche adaptation. Interestingly, 153 orthogroups (including 156 genes) were found exclusively in *D. amygdali* isolates. These included genes encoding putative ABC multidrug transporters, amino acid metabolism enzymes, transcription factors, oxidoreductases, permeases, and phosphoesterases (Appendix A), which may contribute to *D. amygdali*’s unique pathogenic profile or ecological adaptation. Notably, no singleton genes (genes not assigned to any orthogroup) were identified for the DA-1 isolate, indicating that all annotated genes could be grouped with homologs across the *Diaporthe* dataset. In the Orthofinder species tree, the *D. amygdali* isolates clustered together, in a separated branch close to *D. australafricana* and *D. ilicicola* (Appendix A).

### 2.7. Effectors and Virulence-Related Features Among D. amygdali Isolates

A deeper comparison of *D. amygdali* isolates (DA-1, DUCC-20226, and CAA958) has been carried out by looking at both the Average Nucleotide Identity (ANI) and shared orthologous proteins, the carbohydrate-active enzymes (CAZymes), the secreted proteins with effector functions and virulence-associated features according to the PHI-base (Pathogen–Host Interactions) database. Overall, DA-1 from peach shares 99.26% with CAA958 from blueberry and slightly lower (99.16%) with DUCC20226 from apple (Appendix A). A second, *D. amygdali*-specific, Orthofinder analysis revealed 11,984 orthogroups shared among the three isolates, with 5 orthogroups specific for DA-1, including a total of 16 genes related to NADH-dehydrogenase subunits and several hypothetical proteins (Appendix A). The higher level of identity at both nucleotide and orthologous level was further confirmed when comparing the virulence-related features. The CAZyme distribution showed that, in DA-1, the most abundant CAZyme families included 407 glycoside hydrolases (GHs), which cleave glycosidic bonds in saccharides; 237 auxiliary activity (AA) enzymes, often involved in lignin and biomass degradation; and 61 carbohydrate esterases (CEs), which remove ester-linked substitutions from complex carbohydrates. In addition, 111 glycosyltransferases (GTs) involved in polysaccharide biosynthesis, 9 carbohydrate-binding modules (CBMs) that enhance substrate binding, and 34 polysaccharide lyases (PLs) that cleave glycosidic bonds via β-elimination were also identified (Figure 5A). This enzymatic repertoire highlights the pathogen’s capacity to degrade a broad range of plant-derived carbohydrates [33], and results to be comparable to those from blueberry and apple (Figure 5A). The EffectorP prediction on proteins with a signal peptide revealed a total of 493 effectors in DA-1, 448 effectors in DUCC20226 and 513 in CAA958, either apoplastic, cytoplasmatic or a combination of both (Figure 5B).

Annotation using PHI-base (Pathogen–Host Interactions database) identified a total of 4670 virulence-associated features. The majority were associated with reduced virulence or unaffected pathogenicity, followed by genes linked to loss of pathogenicity (Figure 5C). This integrative approach—combining secretion signal prediction, effector likelihood, and pathogenicity annotations—greatly enhances confidence in effector identification. It highlights the complex infection strategy of *D. amygdali* isolates, which likely involves not only canonical isolate-specific effectors but also a broader secretome contributing to host manipulation, immune suppression, and nutrient acquisition during infection.

### 2.8. Fusicoccin Biosynthetic Gene Cluster

A particularly notable finding is the identification of a complete fusicoccin biosynthetic gene cluster (BGC) on Contig 2 of the *D. amygdali* DA-1 genome (Figure 6). Fusicoccin is a well-characterized diterpenoid phytotoxin, originally isolated from *Phomopsis amygdali*, and its detection here suggests a conserved pathogenic mechanism in *D. amygdali* [35]. The cluster encodes the central enzyme fusicoccadiene synthase (PaFS), a bifunctional diterpene synthase featuring a C-terminal prenyltransferase domain (for GGDP synthesis) and an N-terminal cyclase domain that produces fusicoccadiene, the tricyclic precursor of fusicoccin A [36,37]. PaFS is known to form multimeric enzyme complexes, enabling efficient substrate channeling during biosynthesis [37]. In addition to PaFS, the cluster includes cytochrome P450 monooxygenases, a dioxygenase, and an O-prenyltransferase (PAPT), all involved in subsequent tailoring reactions that generate the structural diversity of fusicoccin A [35]. These genes show high sequence similarity to those of the canonical fusicoccin pathway, underscoring *D. amygdali*’s capacity to synthesize biologically active fusicoccin or analogs [35,36,37,38]. Although these genes do not encode typical protein effectors, their products act as non-proteinaceous virulence factors, capable of profoundly affecting host physiology. Specifically, fusicoccin irreversibly activates plasma membrane H^+^-ATPases in guard cells, leading to stomatal opening, excessive water loss, and tissue collapse. Interestingly, while previous studies [38,39] reported the fusicoccin cluster across two separate loci, all thirteen genes of the cluster were found in a single contiguous region on the *D. amygdali* CAA958 isolate (GenBank accession NW_026517024, NODE4, positions 1,044,299–1,085,864). When this region was blasted against both DA-1 and DUCC20226 isolates, it revealed 99.9% sequence identity within a single locus on Contig 2 in the first one (Figure 6), and 97% in the latter, confirming both sequence conservation and physical clustering. Additionally, OrthoFinder analysis identified a second, shorter PaFS homolog (0G117690) on Contig 7, sharing only 44% amino acid identity with the cluster-associated PaFS. However, no other genes from the biosynthetic pathway were present at this locus, suggesting a possible pseudogene or a paralog with a different functional context.

## 3. Discussion

Traditional diagnostic methods, often based on morphological traits or biochemical assays, are limited in their ability to distinguish closely related or cryptic species, potentially resulting in misdiagnosis and ineffective control measures [40]. In contrast, the integration of genomic data into epidemiological surveillance enables precise tracking of pathogen origin, spread, and evolution [28]. Moreover, genomic resources underpin the development of rapid, sensitive, and specific molecular diagnostics for early detection and real-time monitoring in agricultural settings [21,22,24,25,27,41,42].

This study presents the first comprehensive genomic characterization of *D. amygdali* isolated from *Prunus persica* twigs, a re-emerging threat in European peach orchards that can dramatically reduce yield and shorten plant lifespan, thereby increasing production costs [4]. The high-quality genome assembly of *D. amygdali* DA-1 provides a comprehensive foundation for understanding the genetic basis of pathogenicity and host adaptation. The combination of long-read PacBio HiFi and Illumina short-read sequencing enabled the generation of a near-complete genome assembly, with most chromosomes fully or nearly fully resolved. This chromosomal-level assembly facilitates detailed structural and functional genomic analyses that were previously limited in this genus.

The genome’s compartmentalization into AT-rich and GC-rich regions, with clear signatures of Repeat-Induced Point mutation (RIP) activity in the former, reflects a common fungal genome defense strategy to limit transposable element proliferation [31]. The AT-rich regions are gene-poor and likely represent genomic “repeat deserts” shaped by RIP, while the GC-rich compartments maintain gene density and functional content. Notably, the lack of identifiable *rid* homologs raises intriguing questions about the molecular basis and evolutionary history of RIP-like processes in *D. amygdali*, warranting further investigation [32].

The high completeness of the genome assembly (BUSCO 99.2%) and the extensive functional annotation achieved across multiple databases support the reliability of this resource for downstream applications. The presence of numerous carbohydrate-active enzymes (CAZymes), membrane transporters, and cytochrome P450s underscores the pathogen’s capacity for host invasion, nutrient acquisition, and secondary metabolite production [33]. The identification of a broad repertoire of secondary metabolite biosynthetic gene clusters, including those for known phytotoxins and potential novel compounds, together with a large set of candidate effectors, highlights the metabolic versatility and the complexity of the secretome of *D. amygdali.*

Taken together, these findings suggest a coordinated modulation of host immunity, suppression of defenses, exploitation of host resources and competition within the plant microbiome [43].

A particularly significant finding is the identification of a complete and physically clustered fusicoccin biosynthetic gene cluster (BGC). This cluster suggests the production of fusicoccin, a diterpenoid phytotoxin essential for *D. amygdali* virulence through irreversible activation of host plasma membrane H^+^-ATPases, leading to stomatal opening, excessive water loss, and tissue collapse [35,36,37,38,39]. The conservation and clustering of these genes within a single genomic locus highlight a potential regulatory hub critical for toxin biosynthesis. Importantly, this cluster represents a promising molecular target for novel disease management approaches. Specifically, the development of double-stranded RNA (dsRNA)-based gene silencing strategies targeting key enzymes within the fusicoccin pathway, such as the central synthase PaFS, could effectively reduce virulence by suppressing toxin production. This approach aligns with sustainable plant protection goals by minimizing reliance on chemical fungicides [44].

Comparative genomic analyses positioned DA-1 within the broader *Diaporthe* genus, revealing both core orthogroups shared across species and lineage-specific expansions that may contribute to host adaptation. Comparison among the three available *D. amygdali* genomes—DA-1 from peach, CAA958 from blueberry, and DUCC20226 from apple—showed consistently high genomic relatedness, with ANI values exceeding 99%. Despite the overall conservation, each isolate displayed a small complement of unique orthogroups, including genes encoding hypothetical proteins, metabolic enzymes, and putative virulence factors, which may underpin subtle differences in host interaction. Effector prediction further revealed quantitative variation, with DA-1 encoding 493 candidates compared to 448 in DUCC20226 and 513 in CAA958, indicating both shared core effectors and isolate-specific expansions. CAZyme repertoires were also broadly similar in family distribution, reflecting a conserved ability to degrade plant polysaccharides, while minor differences may relate to host specialization. Taken together, these observations suggest that while *D. amygdali* maintains a stable core genome across hosts, modest isolate-specific genomic features could contribute to adaptation to distinct plant environments.

Together, these genomic insights provide a detailed molecular framework for *D. amygdali* biology and pathogenicity. Future functional studies leveraging this resource will be critical to unraveling the regulatory mechanisms controlling virulence factor expression and to identifying novel targets for disease control. The genomic data also offer opportunities for comparative analyses across related pathogens, advancing our understanding of fungal evolution, host specialization, and the molecular arms race between plants and their pathogens. Overall, the high genomic similarity among *D. amygdali* isolates from peach, apple, and blueberry, coupled with isolate-specific effector and orthogroup variation, points to a stable core genome with fine-scale adaptations that may shape host specificity. The integration of high-quality genome sequencing, functional annotation, and comparative analysis thus offers a powerful platform to better understand and address *D. amygdali*, thereby safeguarding peach production in Italy and globally in line with the objectives of sustainable agriculture and food security.

## 4. Materials and Methods

### 4.1. Isolate Selection and HMW DNA Extraction

Following a field survey carried out in Emilia Romagna in April 2024, several *D. amygdali* isolates have been morphologically and molecularly characterized as described in Brugneti et al. [4]. Whole-genome sequencing and downstream analysis were performed on *D. amygdali* DA-1, which was isolated from a cirrus that was perfectly visible under the microscope [4]. Monohyphal cultures of the DA-1 isolate were initially grown on water agar (WA) and subsequently transferred to potato dextrose agar (PDA), followed by incubation at 25 °C. After one week, a mycelial plug was inoculated into 250 mL of potato dextrose broth (PDB) and incubated in a rotary shaker at 150 rpm for 10 days at 25 °C. The fresh mycelium was filtered and dried at room temperature under laminar flow for 30 min. Two grams of mycelium were ground in liquid nitrogen, transferred into a new 50 mL tube and incubated for 30 min at 65 °C with 15 mL of CTAB-based extraction buffer (2% CTAB, 2 mM EDTA [pH 8.0], 100 mM TrisHCl [pH 8.0], 1.2 M NaCl) and 200 µL proteinase K (20 mg/mL). Purification was performed twice with one volume of chloroform:isoamyl alcohol (24:1), followed by centrifugation at 5000× *g* for 10 min to not damage the HMW DNA. The clear aqueous phase was transferred into a new 50 mL tube and incubated with 10% CTAB buffer pre-heated at 65 °C, together with one volume of chloroform:isoamyl alcohol (24:1). After centrifugation, the supernatant was transferred into a new 50 mL tube adding ⅔ volume of cold (−20 °C) isopropanol and 10% (*v*/*v*) of 5M sodium acetate, and then incubated at −20 °C for 2 h. Precipitated DNA was recovered by a centrifugation step at 5000× *g* for 20 min, and the pellet was washed twice with 1 mL of pure ethanol. The pellet was then dried under the laminar flow, resuspended in 500 µL of preheated (65 °C) Tris-EDTA (TE) buffer, and incubated at 37 °C for 30 min. The integrity of high-molecular-weight (HMW) DNA was assessed by 1% agarose gel electrophoresis, while concentration was measured using a Qubit fluorometer (Thermo Fisher Scientific, Waltham, MA, USA), and purity was evaluated with an OPTIZEN NanoQ spectrophotometer (KLAB, Yuseong-gu, Daejeon, Republic of Korea).

### 4.2. Whole-Genome Sequencing and Assembly

Genome sequencing was performed from a total of 7.5 μg of HWM DNA by Biomarker Technologies (BMK) GmbH (Münster, Germany) with Illumina NovaSeq 6000 (Illumina, San Diego, CA, USA) and PacBio HiFi (CCS) platforms according to their internal procedures for library preparation and sequencing. After quality check assessment using FastQC v0.11.8 [45], the ccs reads were assembled by Hifiasm software v0.20 [46] and further polished with the short Illumina reads using BWA v0.7.12 [47] and Pilon v1.23 [48]. The mitochondrial genome was assembled with MitoHiFi v3.2.2 [49], using *D. eres* ZM79-3 (Accession Number PQ493439.1) and *D. longicolla* TWH P74 (Accession number PP820856.1) as references. Genome assembly quality statistics were evaluated using QUAST v5.0.2 [50], and genome coverage was retrieved by reads alignment through BWA, followed by Samtools v1.13 [51] and the *bamqc* function of the *qualimap* tool [52]. BUSCO v5.beta.1 [53] was employed to assess the integrity of the fungal genome assembly, using the 290 conserved core genes of the fungi_odb9 as an ortholog lineage dataset. A graphical representation of the assembled genome has been created using Circos v0.69-8 [54].

### 4.3. Structural and Functional Annotation

Transposable elements were annotated using HiTE [55]. Once the repeats have been masked, Genscan [56], Augustus v2.4 [57], GlimmerHMM v3.0.4 [58], GeneID v1.4 [59], SNAP v2006-07-28 [60] for de novo prediction. GeMoMa v1.3.1 [61] was used for homologous protein prediction, while EVM v1.1.1 [62] was finally used to combine the prediction results obtained by the above different algorithms. tRNAs were predicted with tRNAscan-SE v2.0.12 [63], while ncRNAs and rRNAs were predicted using Infernal v1.1 and Rfam database, respectively [64,65]. OcculterCut v1.1 was employed to detect AT-rich isochores, while RIPCAL v1.0 was used to calculate dinucleotide ratios indicative of potential RIP activity [66,67].

The predicted gene sequences were aligned with functional databases such as KOG [68], KEGG [69], BLASTp, and the latest version of the SwissProt 2023_05 database. TrEMBL [70], Pfam [71], and Blast2GO [72] were employed for the biological classification.

Biosynthetic Gene Clusters (BGCs) involved in the production of secondary metabolites were automatically searched and analyzed by AntiSMASH v6.0 [73]. Prediction of transmembrane transport was performed with the Transporter Classification database (TCDB, [74]), while signal peptides were identified with SignalP v6 [75]. Given their already known involvement in pathogen–host interaction, the cytochrome P450 enzymes were searched on the Cytochrome P450 engineering database (CYPED, [76]).

### 4.4. Pathogenicity-Related Features Among D. amygdali Isolates

A deeper comparative analysis was performed among the three annotated *D. amygdali* isolates: DA-1, CAA958 from blueberry and DUCC20226 from apple. At the genome level, Average Nucleotide Identity (ANI) was calculated with the *pyani* script using the MUMMER algorithm [77]. The carbohydrate-active enzymes (CAZymes) involved in carbohydrate metabolism were identified through the dbCAN2 meta web server [78]. Potential effector proteins were identified among the proteins containing a signal peptide throughout EffectorP [79]. Pathogenicity and virulence-related features were identified through the Pathogen Host Interactions (PHI) database [80]. Due to its importance in the *Diaporthe* virulence, the fusicoccin-A gene cluster was retrieved from the *D. amygdali* CAA958 genome [38] and searched on the DA-1 and DUCC20226 isolates through BLASTn and BLASTp alignment.

### 4.5. Ortholog Identification and Phylogenetic Analysis

For the comparative genomics analysis, all available *Diaporthe* genome annotations were downloaded from the NCBI Genome database (Table 2). Orthologous proteins were identified using OrthoFinder v2.5.5 [81], and the results were processed further using the R package UpsetR v1.4 [82] within the R environment (v4.2.3). The species tree built by OrthoFinder was visualized in a dendrogram using FigTree v1.4.4 (http://tree.bio.ed.ac.uk/software/figtree/, accessed on 6 August 2025) and further edited with Inkscape v0.92 (https://inkscape.org, accessed on 7 August 2025).

## Figures and Tables

**Figure 1 plants-14-02960-f001:**
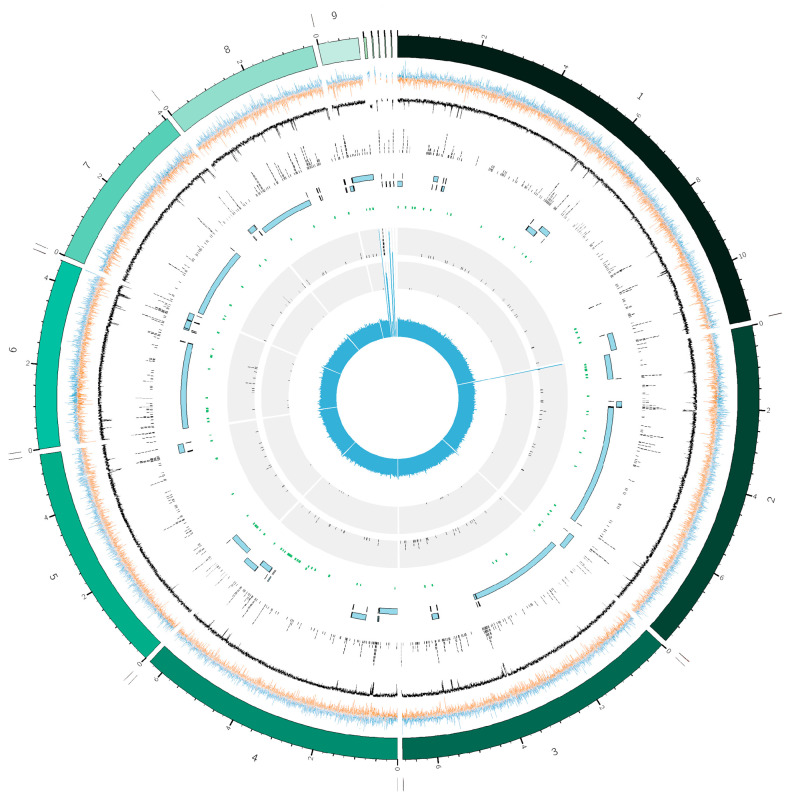
Circos graphical representation of the assembled *Diaporthe amygdali* DA-1 genome. Concentric circles, from outermost to innermost, show: nine main contigs (from 1 to 9), with telomeres indicated as vertical bars when present; additional 5 contigs representing the complete mitochondrial genome, rRNA gene cluster and transposons; GC skew, in blue for the forward strand and orange for the reverse strand; GC content percentage; Repetitive regions; Occultercut R1 region (in light blue) with an equilibrated GC content and the R0 region (in black) enriched in AT; AntiSMASH Gene clusters; tRNAs, rRNAs, Illumina short reads coverage on the assembled sequences.

**Figure 2 plants-14-02960-f002:**
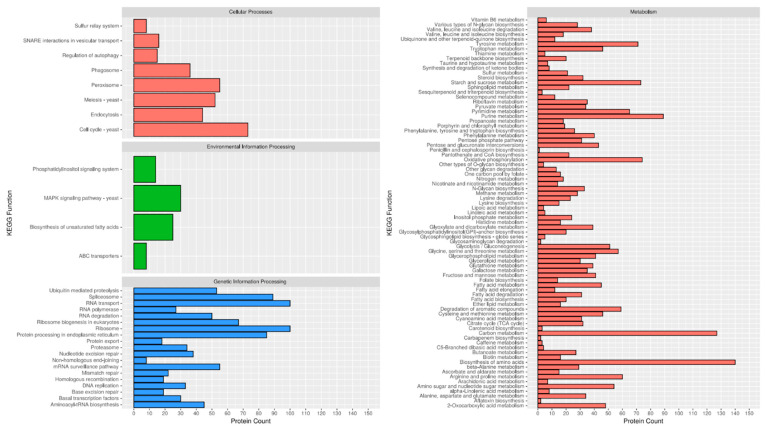
KEGG pathway analysis of *Diaporthe amygdali* genome and distribution among the different three main classes: Genetic Information Processing, Environmental Information Processing, Metabolism, and Cellular Processes.

**Figure 3 plants-14-02960-f003:**
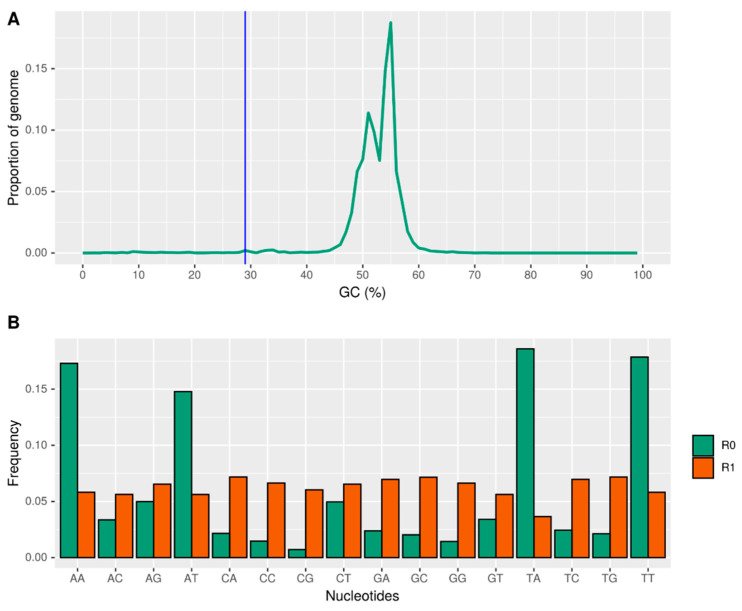
*Diaporthe amygdali* genome shows a bimodal GC content with AT-rich (R0) and GC-rich (R1) regions (**A**). R0 regions are gene-poor and enriched in RIP-associated dinucleotides (AA, AT, TT, TA) compared to R1 (**B**), indicating RIP activity.

**Figure 4 plants-14-02960-f004:**
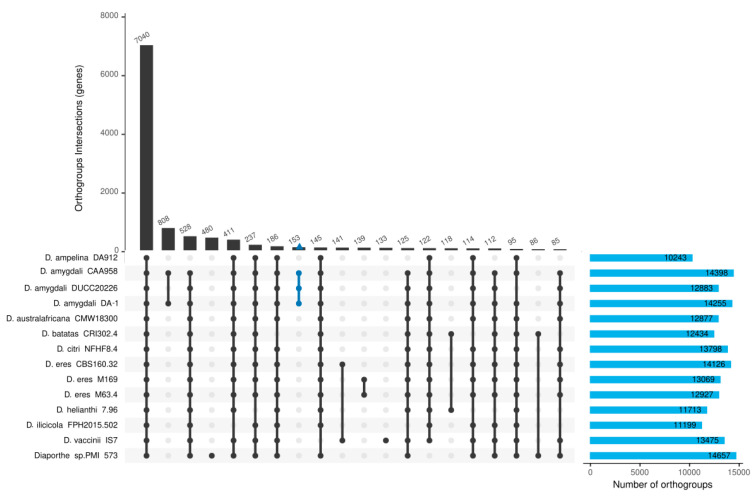
UpsetR plot showing the number of orthogroups identified by OrthoFinder and shared among *Diaporthe amygdali* DA-1 and 13 related isolates, indicated as dots connected by vertical bars. The horizontal blue bars on the right indicate the number of orthogroups per each isolate. The blue vertical line represents the orthogroups shared among the three *D. amygdali* isolates.

**Figure 5 plants-14-02960-f005:**
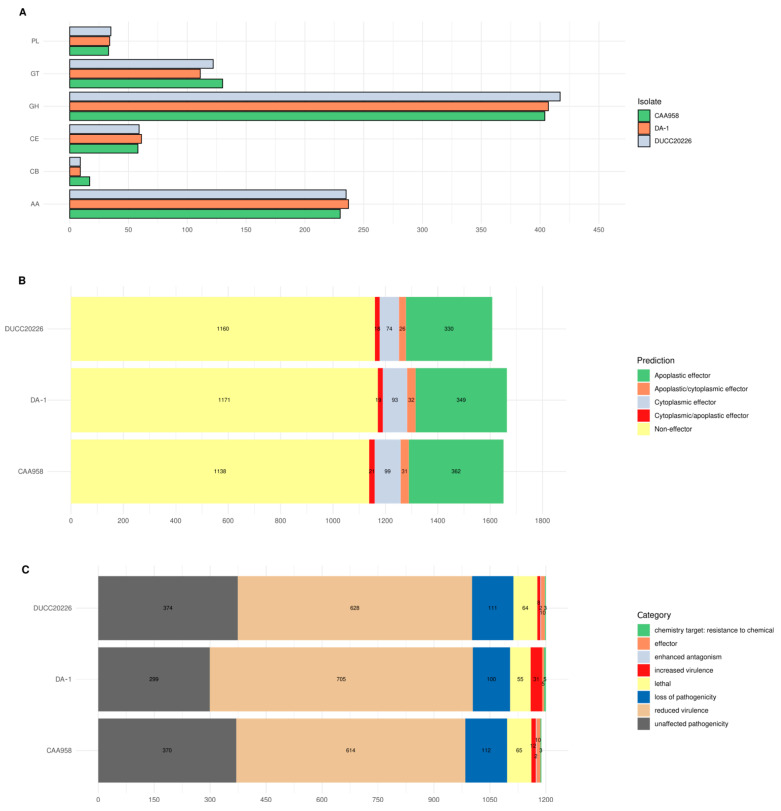
Comparative analysis among *D. amygdali* isolates. (**A**) Distribution of carbohydrate-active enzyme (CAZyme) families, highlighting the abundance of glycoside hydrolases (GHs), auxiliary activity enzymes (AAs), carbohydrate esterases (CEs), glycosyltransferases (GTs), carbohydrate-binding modules (CBMs), and polysaccharide lyases (Pls); (**B**) Effector prediction on proteins with secretion signal; (**C**) Putative pathogenicity-related (PHI-base) features identified by BLASTp analysis of the annotated proteins towards the PHI database.

**Figure 6 plants-14-02960-f006:**
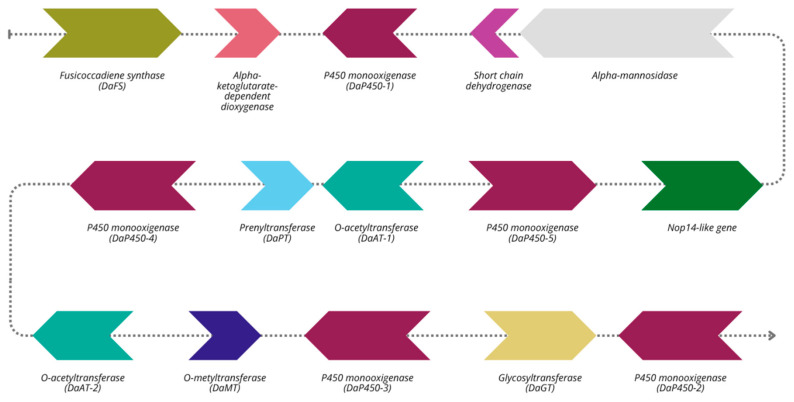
Complete scheme of fusicoccin biosynthetic gene cluster (BGC). Dotted lines represent the genomic sequence; color-coded arrows represent the predicted functions of different genes and indicate their transcriptional direction.

**Table 1 plants-14-02960-t001:** *Diaporthe amygdali* DA-1 assembly statistics.

Genomic Metrics	*D. amygdali* DA-1
# Contigs	14
Genome size	51,431,816
Largest Contig	11,470,755
N50	6,815,508
N90	4,044,788
L50	3
L90	7
GC content %	52.64
# N’s per 100 kbp	0
# N’s	0
Genome coverage	76.42x
BUSCO completeness	99.2%
Complete	253
Single copy	253
Duplicated	0
Fragmented	2
Missing	0
Predicted genes	15,206
rRNAs	51
tRNAs	160
ncRNAs	84
Signal peptide	1799
Retroelements	663
DNA transposons	37
Simple repeats	11,811

**Table 2 plants-14-02960-t002:** List of *Diaporthe* isolates used for comparative analysis.

Accession Number	Organism Name	Strain	Host
GCA_026229845.1	*Diaporthe amygdali*	CAA958	*Vaccinium corymbosum*
GCA_019321695.1	*Diaporthe batatas*	CRI 302-4	*Ipomoea batatas*
GCA_014595645.1	*Diaporthe citri*	NFHF-8-4	*Citrus reticulata* cv. ‘Nanfengmiju’
GCA_023242295.1	*Diaporthe ilicicola*	FPH2015-502	*Ilex verticillata*
GCA_001006365.1	*Diaporthe ampelina*	DA912	*Vitis vinifera*
GCA_032433605.1	*Diaporthe amygdali*	DUCC20226	*Apple tree*
GCA_022570775.2	*Diaporthe eres*	M63-4	*Malus* sp.
GCA_022570805.2	*Diaporthe eres*	M169	*Prunus armeniaca*
GCA_024867555.1	*Diaporthe eres*	CBS 160.32	*Vaccinium corymbosum*
GCA_021399435.1	*Diaporthe* sp.	PMI_573	not applicable
GCA_042826995.1	*Diaporthe vaccinii*	IS7	*Vaccinium macrocarpon* cv. ‘Stevens’
GCA_001702395.2	*Diaporthe helianthi*	Jul-96	*Helianthus* sp.
GCA_042257625.1	*Diaporthe australafricana*	CMW 18300	*Elegia capensis*

## Data Availability

The genome assembled in this work is available on the NCBI Genome database under the accession number JBPXNF000000000.

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
