# Peer review of "Harnessing Genomics of Diaporthe amygdali for Improved Control of Peach Twig Canker and Shoot Blight (TCSB)"

_plants, 2025, doi:10.3390/plants14192960_

Round 1

Reviewer 1 Report

Comments and Suggestions for Authors

The manuscript titled “Harnessing Genomics of Diaporthe amygdali for Improved Control of Peach Twig Canker and Shoot Blight (TCSB)” presents the whole-genome sequencing of the Diaporthe amygdali strain DA-1, combining Illumina short-read and PacBio long-read technologies, with the aim of generating the first high-quality genome assembly of D. amygdali isolated from peach in Italy. This work presents a continuation of the previous study: Brugneti, F.; Rossini, L.; Cirilli, M.; Nardecchia, A.; Onofri, M.; Mazzaglia, A.; Turco, S. Elucidation of ‘Twig Canker and Shoot Blight’ (TCSB) in Peach Caused by Diaporthe amygdali in the North of Italy in Emilia-Romagna. Physiol. Plant. (in press). https://doi.org/10.1111/ppl.70428.
The paper is concise, well written, and provides novel data on candidate virulence genes and other factors involved in the pathogenicity of D. amygdali.
I have a minor but important remarks: 
-    Figure 8 in the manuscript is identical to Figure 6(D) from the work by Brugneti, F.; Rossini, L.; Cirilli, M.; Nardecchia, A.; Onofri, M.; Mazzaglia, A.; Turco, S. Elucidation of ‘Twig Canker and Shoot Blight’ (TCSB) in Peach Caused by Diaporthe amygdali in the North of Italy in Emilia-Romagna. Physiol. Plant. (in press). https://doi.org/10.1111/ppl.70428. This constitutes self-plagiarism. The authors should either remove it or replace it with an other original figure.
-    Taking into account the content of the paper and the presented results, i.e., that the authors worked with only one strain and the study focused solely on genome analysis, I suggest publishing paper as type “Communication.”

Author Response

The manuscript titled “Harnessing Genomics of Diaporthe amygdali for Improved Control of Peach Twig Canker and Shoot Blight (TCSB)” presents the whole-genome sequencing of the Diaporthe amygdali strain DA-1, combining Illumina short-read and PacBio long-read technologies, with the aim of generating the first high-quality genome assembly of D. amygdali isolated from peach in Italy. This work presents a continuation of the previous study: Brugneti, F.; Rossini, L.; Cirilli, M.; Nardecchia, A.; Onofri, M.; Mazzaglia, A.; Turco, S. Elucidation of ‘Twig Canker and Shoot Blight’ (TCSB) in Peach Caused by Diaporthe amygdali in the North of Italy in Emilia-Romagna. Physiol. Plant. (in press). https://doi.org/10.1111/ppl.70428.

The paper is concise, well written, and provides novel data on candidate virulence genes and other factors involved in the pathogenicity of D. amygdali.

I have a minor but important remarks: 

-    Figure 8 in the manuscript is identical to Figure 6(D) from the work by Brugneti, F.; Rossini, L.; Cirilli, M.; Nardecchia, A.; Onofri, M.; Mazzaglia, A.; Turco, S. Elucidation of ‘Twig Canker and Shoot Blight’ (TCSB) in Peach Caused by Diaporthe amygdali in the North of Italy in Emilia-Romagna. Physiol. Plant. (in press). https://doi.org/10.1111/ppl.70428. This constitutes self-plagiarism. The authors should either remove it or replace it with an other original figure.

Thank you for pointing this out. The figure 8 has been removed and replaced with a clear citation to Brugneti et al., 2025

-    Taking into account the content of the paper and the presented results, i.e., that the authors worked with only one strain and the study focused solely on genome analysis, I suggest publishing paper as type “Communication.”

We do understand your point but “Communication” is usually a really short text with a few analyses. Given the several and deeper analysis, we would rather keep it as a Full article.

Reviewer 2 Report

Comments and Suggestions for Authors

The paper entitled "Harnessing genomics of Diaporthe amygdali for improved control of peach Twig Canker and Shoot Blight (TCSB)" is generally well written and scientifically important. However, the paper can be improved by:

  1. The authors need to make sure that the Aims are clearly written.
  2. The figures should be approved. For example Figure 1 is almost impossible to read.
  3. Maybe Table 1 can move to the Supplementary tables?
  4. However, my main concern is that the main findings are not compared to other studies.

Author Response

The paper entitled "Harnessing genomics of Diaporthe amygdali for improved control of peach Twig Canker and Shoot Blight (TCSB)" is generally well written and scientifically important. However, the paper can be improved by:

  1. The authors need to make sure that the Aims are clearly written

The aims of the study are reported throughout the main text. The whole genome sequence of a plant pathogen could be of relevant importance to develop future molecular-precise control strategies (which go beyond this manuscript). This is further discussed at the beginning of the Discussion, with additional references added according to revisor 3.  

  1. The figures should be approved. For example Figure 1 is almost impossible to read.

We do agree that the Figures embedded in the main text are difficult to read. But they were included in the main text just to make the peer review process easier. We did upload the figure as separated files as well with much higher resolution so in the possible final publication it would be of better quality

  1. Maybe Table 1 can move to the Supplementary tables?

Table 1 provides important information regarding the overall statistics and quality of the assembled genome. We would like to keep it as a main table. Thanks

  1. However, my main concern is that the main findings are not compared to other studies.

Thanks for pointing this out. Since it’s a Whole genome sequencing Manuscript the reasonable comparison is towards other closely related genomes. This comparison has been done towards 13 more Diaporthe isolates and, recently within this revision process, even deeper towards the 2 other D. amygdali isolates. Please have a look to the improved manuscript.

Reviewer 3 Report

Comments and Suggestions for Authors

The manuscript demonstrates how one can integrate genomic information of a pathogen to improve our understanding of plant-pathogen interactions.   The authors clearly defined the goals and summarized the findings of their research, which integrates molecular biology, pathogen’s genomics, and functional annotations for improved understanding of D. amygdali pathogenic traits.   

The authors expressed how the identification of putative genes involved with pathogenicity leads to the potential use of cutting-edge technology such as SIGS; an approach that, at its infancy, it is innovative. Coupling SIGS and genomic data for both the pathogen and its hosts will result in an improved version of the current integrative management strategies against phytopathogens.

Thank you for providing a well-written manuscript in a scientific research field that could be intimidating to some that are not familiar with molecular characterization of pathogens. Your contribution clearly provides new insights into the role of genes essential for virulence and pathogenicity, improving our understanding of complex and diverse genera like Diaporthe.

Author Response

The manuscript demonstrates how one can integrate genomic information of a pathogen to improve our understanding of plant-pathogen interactions.   The authors clearly defined the goals and summarized the findings of their research, which integrates molecular biology, pathogen’s genomics, and functional annotations for improved understanding of D. amygdali pathogenic traits.   

The authors expressed how the identification of putative genes involved with pathogenicity leads to the potential use of cutting-edge technology such as SIGS; an approach that, at its infancy, it is innovative. Coupling SIGS and genomic data for both the pathogen and its hosts will result in an improved version of the current integrative management strategies against phytopathogens.

Thank you for providing a well-written manuscript in a scientific research field that could be intimidating to some that are not familiar with molecular characterization of pathogens. Your contribution clearly provides new insights into the role of genes essential for virulence and pathogenicity, improving our understanding of complex and diverse genera like Diaporthe.

Introduction 

This is a very well-written introduction including significance and justification for their research as well as the application and impact of their findings. Thank you. 

Thank you very much for these nice comments. Getting our work really appreciated is the best result we could obtain. 

Results 

Lines 144 and 149. Correct figure number within parentheses; change Figure 4A and 4B to Figure 3. 

The figure number has been corrected, thanks for pointing it out. 

Line 251. Change the figure number from Figure 5 to Figure 6. 

All the figures have been adjusted, thanks.

Discussion 

Lines 306 through 311. As presented, it is not clear how genomics are used to improve diagnostics. It would be quite difficult for inexperienced scientists in molecular biology and genomics to know how the information and data presented here will assist them on diagnosis of TCSB. Suggestion, please clarify. 

Thanks for this suggestion. We believe that pointing at the relevant literature you suggested already provides necessary information. 

Materials and Methods 

Line 457. Fix indentation of subheading 4.5 

Done thanks.

Consider adding review articles of general concepts such as the ones included below. 

He, L., Zhou Y., Mo, Q., Huang, Y., and Xueming Tang, X. (2024). Spray-induced gene silencing in phytopathogen: Mechanisms, applications, and progress. Advanced Agrochem 3(4):289-297. https://doi.org/10.1016/j.aac.2024.06.001. 

Klosterman S. J., Rollins, J. R., Sudarshana, M. R., and Vinatzer, B. A. 2016. Disease Management in the Genomics Era—Summaries of Focus Issue Papers. Phytopathology 106:10, 1068-1070. https://doi.org/10.1094/PHYTO-07-16-0276-FI 

Sarethy, I.P., Saharan, A. Genomics, proteomics and transcriptomics in the biological control of plant pathogens: a review. 2021. Indian Phytopathology 74, 3–12. https://doi.org/10.1007/s42360-020-00302-2 

Wilson, H., Bar, I., Hobson, K., Vaghefi, N., Petronitis, T. and Ford, R. (2025), Integrated Disease Management, Adaptation and Genomics of Fungal Plant Pathogens in Cropping Systems. Plant Pathol, 74: 1445-1469. https://doi.org/10.1111/ppa.14114

Thank you very much for suggesting these interesting papers. They have been added.

Reviewer 4 Report

Comments and Suggestions for Authors

This study aims to elucidate the genomic information of D. amygdali in order to understand the pathogenicity for peach twig canker and shoot blight. Based on the high completeness of the genome assembly and gene predictions, it is considered that reliable genomic information for this species has been obtained. However, compared to previous genome analyses of this species, novel information is extremely limited.

In the abstract, the authors emphasized the importance of understanding this species' pathogenicity and how it uses pathogenicity-related genes, such as spray-induced gene silencing. However, this perspective is not related to the data shown in this study. The focus should be on the novel results of this study compared to previous studies, not on future prospects far removed from the presented results.

 Hilário et al. (2022) presented genome data for Diaporthe amygdali isolated from blueberries and performed comparative genome analyses among various Diaporthe species. This study differs in that it uses an isolate from peach. If any novel findings are found in the pathogeneic-related genes of D. amygdali, the comparative genomic analyses should focus on the differences found in the peach isolate's genome and/or the novel findings added by the peach isolate to the analyses in this study. It is also important to compare this study with Mena et al.'s (2022) paper on the comparative genomics of D. caulivora.

Analyses of orthologs specific to D. amygdali might be related to the pathogenicity of this species; however, the analyses of the genes are superficial and show limited information. In a scientific paper, you should present any novel findings from your analyses.

L365-376: The authors argue that the fusicoccin biosynthesis cluster is a regulator of P. amygdali pathogenicity. However, there is no experimental evidence that fusicoccin plays a major role in pathogenicity of D. amygdal. It was only recognized that fusicoccin biosynthesis cluster is present in a chromosome of the D. amygdali isolate from peach, in addition to blueberry isolates shown in the previous study. Therefore, they should rewrite content open to debate in scientific significance while referencing prior studies.

L272-273, L457-: Ibid as above.

L361-, L457- : Did you check the pathogenicity of DA-1 isolate on peach twig?

Author Response

This study aims to elucidate the genomic information of D. amygdali in order to understand the pathogenicity for peach twig canker and shoot blight. Based on the high completeness of the genome assembly and gene predictions, it is considered that reliable genomic information for this species has been obtained. However, compared to previous genome analyses of this species, novel information is extremely limited.

Thank you for your consideration. The novelty in this paper is that this is the first genome of a Diaporthe amygdali from peach, which resulted to be quite conserved with the only other two genomes sequenced and annotated. We do believe that sharing high-quality genetic information with the scientific community (i.e. public database) is fundamental for future progress. More genomes of this species are needed for a broader and deeper comparative analysis in order to find the EUREKA we are all looking for.  

In the abstract, the authors emphasized the importance of understanding this species' pathogenicity and how it uses pathogenicity-related genes, such as spray-induced gene silencing. However, this perspective is not related to the data shown in this study. The focus should be on the novel results of this study compared to previous studies, not on future prospects far removed from the presented results.

We have removed this mention to the SIGS methods, since it is too speculative and related to future studies. However, genetic information are necessary to develop precise diagnostic and control strategies, we just wanted to mention this. 

Hilário et al. (2022) presented genome data for Diaporthe amygdali isolated from blueberries and performed comparative genome analyses among various Diaporthe species. This study differs in that it uses an isolate from peach. If any novel findings are found in the pathogeneic-related genes of D. amygdali, the comparative genomic analyses should focus on the differences found in the peach isolate's genome and/or the novel findings added by the peach isolate to the analyses in this study. 

We added a deeper comparative analysis between the three available D. amygdali isolates (from peach, blueberry and apple). Differences rely only on a few genes (16 genes belonging to 5 orthogroups) which functional information is missing. The isolates share above 99% of identity and more than 11k orthogroups. More than differences, we could also highlight how these isolates are similar to each other, even in the fusicoccin gene cluster structure. Yet, as mentioned before, genetic variability could arise from a broader investigation, thus we need more genome sequenced!  

It is also important to compare this study with Mena et al.'s (2022) paper on the comparative genomics of D. caulivora.

We did include a comparative analysis between D. amygdali isolates, but we did not include any references from Mena et al., (2022) since nothing has been taken from this paper. Just to be clear, D. caulivora was not included in the comparative analysis because its annotation is not available on the NCBI Genome database. 

 Analyses of orthologs specific to D. amygdali might be related to the pathogenicity of this species; however, the analyses of the genes are superficial and show limited information. In a scientific paper, you should present any novel findings from your analyses.

We believe to have answered this comment in the previous response. 

L365-376: The authors argue that the fusicoccin biosynthesis cluster is a regulator of P. amygdali pathogenicity. However, there is no experimental evidence that fusicoccin plays a major role in pathogenicity of D. amygdal. It was only recognized that fusicoccin biosynthesis cluster is present in a chromosome of the D. amygdali isolate from peach, in addition to blueberry isolates shown in the previous study. Therefore, they should rewrite content open to debate in scientific significance while referencing prior studies.

For decades, fusicoccin has been known to be a toxin involved in fungal pathogenesis, even in species different from Diaporthe. We believe that it would have been out of context (and time!) testing its functional involvement by creating mutant-depleted isolates! Being a whole genome related paper, our main focus was to identify the gene cluster and assess its conserved sequence. The presence of a full gene cluster clearly suggests that the toxin itself could be produced and involved in this isolate’s pathogenicity. This doesn’t mean that this toxin is the ONLY virulence-related feature either. 

L272-273, L457-: Ibid as above

L361-, L457- : Did you check the pathogenicity of DA-1 isolate on peach twig?

Yes we did, in Brugneti et al., 2025, where we isolated and characterized this DA-1 isolate. Please check it : Physiologia Plantarum, 2025; 177:e70428 https://doi.org/10.1111/ppl.70428

Round 2

Reviewer 2 Report

Comments and Suggestions for Authors

The authors have addressed most of my concerns. However, they still, In my opinion, does not included references in the Discussion.

For example, the following paragraphs does not included a single reference. "The genome’s compartmentalization into AT-rich and GC-rich regions, with clear signatures of Repeat-Induced Point mutation (RIP) activity in the former, reflects a common fungal genome defense strategy to limit transposable element proliferation. The AT-rich regions are gene-poor and likely represent genomic “repeat deserts” shaped by RIP, while the GC-rich compartments maintain gene density and functional content. Notably, the lack of identifiable rid homologs raises intriguing questions about the molecular basis and evolutionary history of RIP-like processes in D. amygdali, warranting further investigation." (Lines 336-343).

Author Response

Dear reviewer,

After a careful consideration of the Discussion, we have added more references (including also the ones suggested by reviewer 3) and removed some repetitive sentences to have a more fluid and clear text.

We hope that now the Manuscript fit the journal's style and requirements.

Thanks for your suggestions.